# Comparative Study of the Microstructure and Properties of Cast-Fabricated and 3D-Printed Laser-Sintered Co–Cr Alloys for Removable Partial Denture Frameworks

**DOI:** 10.3390/ma16083267

**Published:** 2023-04-21

**Authors:** Dejan Stamenković, Miljana Popović, Rebeka Rudolf, Milorad Zrilić, Karlo Raić, Kosovka Obradović Đuričić, Dragoslav Stamenković

**Affiliations:** 1Dental Office—Stamenkovic and Team, 11000 Belgrade, Serbia; dr.dejan.stamenkovic@gmail.com; 2Faculty of Technology and Metallurgy, University of Belgrade, 11120 Belgrade, Serbia; miljana@tmf.bg.ac.rs (M.P.); misa@tmf.bg.ac.rs (M.Z.); karlo@tmf.bg.ac.rs (K.R.); 3Faculty of Mechanical Engineering, University of Maribor, 2000 Maribor, Slovenia; 4School of Dental Medicine, University of Belgrade, 11000 Belgrade, Serbia; kosovkaobradovicdjuricic@gmail.com; 5Academy of Medical Science, Serbian Medical Society, 11000 Belgrade, Serbia; d.stamenkovic49@gmail.com

**Keywords:** Co–Cr dental alloys, 3D printing, laser melting and sintering, casting, microstructure, mechanical properties, characterisation

## Abstract

Since additive technologies in dentistry are gradually replacing metal casting technology, it is necessary to evaluate new dental constructions intended for the development of removable partial denture frameworks. The aim of this research was to evaluate the microstructure and mechanical properties of 3D-printed, laser-melted and -sintered Co–Cr alloys, and perform a comparative study with Co–Cr castings for the same dental purposes. The experiments were divided into two groups. The first group consisted of samples produced by conventional casting of the Co–Cr alloy. The second group consisted of 3D-printed, laser-melted and -sintered specimens produced from a Co–Cr alloy powder divided into three subgroups, depending on the technological parameters chosen for manufacturing (angle, location and heat treatment). Examination of the microstructure was carried out by classical metallographic sample preparation, using optical microscopy and scanning electron microscopy with energy dispersive X-ray spectroscopy (EDX) analysis. A structural phase analysis was also performed by XRD. The mechanical properties were determined using a standard tensile test. The microstructure observation showed a dendritic character in the case of castings, while in the case of 3D-printed, laser-melted and -sintered Co–Cr alloys, the microstructure was typical for additive technologies. The XRD phase analysis confirmed the presence of Co–Cr phases (ε and γ). The results of the tensile test showed remarkably higher yield and tensile strength values and slightly lower elongation of the 3D-printed, laser-melted and -sintered samples than those produced by conventional casting.

## 1. Introduction

The therapy of partially edentulous patients with mobile restorations depends, among other things, on the chemical, physical and functional properties of the removable partial denture (RPD) metal framework. Theoretically, the manufacturing of the RPD framework is possible by casting—making a metal framework by replacing a wax model with a metal alloy; by substrate technologies—making a metal framework by cutting material from a solid block of material; and by additive technologies—making a metal framework by adding, applying and depositing material.

Although casting RPD frameworks has been the gold standard for many years, additive technologies are suppressing the use of this technology due to their numerous advantages [1,2,3]. Additive technologies involve the process of joining particles of material in the desired 3D shape. As these technologies do not require making tools or long adjustments to machines, they are called fast or direct production technologies (rapid prototyping—RP) [4]. This technology arose from the need to make prototypes (hence the name) and is very suitable for making small series. In prosthodontics, it is used successfully in the manufacturing of fixed restorations on prepared teeth and implants. Optimal solutions for mobile prosthetics are still lacking.

Additive technologies shorten the preparation time, because the input is a 3D digital geometric model obtained by direct scanning or generation in a commercial CAD program. Of all the additive technologies (stereolithography, selective laser melting and sintering (SLM), direct metal laser sintering (DMLS), 3D printing, fabrication by lamination), the most widely used in dental prosthetics is the selective laser melting and sintering of metal powder particles [5,6]. Additionally, advances in digital technology and imaging over the past 30 years have enabled the implementation of three-dimensional (3D) modelling protocols in dentistry. The use of stereolithographic models gradually replaced the traditional ground models practically [7,8,9].

SLM/DMLS technology in the manufacturing of fixed dental restorations is acceptable, due to the small dimensions and rigidity of all parts of fixed dental restorations [10]. The dimensions of the metal RPD frameworks are significantly larger, and rigidity is required only for the major connector. Depending on the type of partial edentulousness, minor connectors should either be rigid or elastic. Clasp arms are always elastic; therefore, the application of SLM/DMLS technology in the production of metal RPD frameworks is significantly limited [6,11]. The therapy success of partially edentulous patients with RPD depends on the mechanical properties of the alloy defined by its microstructure [11,12,13,14].

On the basis of the above, the primary goal of this study was to compare the microstructure and mechanical properties of 3D-printed, laser-melted and -sintered Co–Cr alloys produced at different technological parameters with castings from the same Co–Cr alloy for similar dental applications. We have assessed that such a concrete comparison is necessary, as it concerns the use of a Co–Cr alloy for prosthetic restorations, where the properties and structure, or the internal structure, are not sufficiently well known to the users. Specifically, any change in the microstructure always results in a change in the final properties of the prosthetic restorations. The resulting properties are highly dependent on the technology of production, processing and procedures with which we build and create prosthetic restorations for the needs of patients in dental practice. Therefore, knowledge of the key properties of Co–Cr alloys is required, depending on the manufacturing technology. The purpose of this article is to demonstrate to the wider scientific community that there are differences in microstructure, and, consequently, in the properties of Co–Cr dental alloys, despite identical raw material output, due to different manufacturing and subsequent heat treatment technology.

## 2. Materials and Methods

### 2.1. Sample Preparation

The nominal chemical compositions of the commercial Co–Cr alloys, which we used for testing in this research, are shown in Table 1. The nominal chemical compositions were provided by the manufacturers Dentaurum GmbH & Co. KG (Ispringen, Germany) and EOS GmbH Electro Optical Systems, Krailling, Germany.

Standard tensile specimens were prepared for testing according to ISO 22674 [15], with dimensions of 3 mm in diameter and a gauge length of 18 mm for both types of Co–Cr alloys. The specimens were named according to the manufacturing technique: conventional casting method (Group C) and DMLS—direct metal laser sintering (Group S).

Tensile specimens prepared by conventional casting (Group C) were used as a reference or control specimens’ state. They were prepared by melting the Co–Cr alloy and pouring it into a mould (Figure 1).

The tensile specimens’ model, which was representative of the casting technique, was designed according to EN ISO 22674: 2022 [12] and printed in a 3D printer (Formlabs Inc., Somerville, MA, USA) in photopolymer (3D Dental, Rapid Model Resin, Monocure, Sydney, Australia). Then, the test specimen models were inserted into the investment materials, and the refractory block was preheated and heated. The Co–Cr alloy was melted in an induction circuit, and casting was performed in a Rotax machine (Fonax T, BEGO, Bremen, Germany) in the presence of air.

The specimens which were representative for the DMLS technique were drawn using CAD/CAM technology according to the dimensions from standard [15], then converted into an STL file and sent to the CNC machine (EOSINT M270, EOS GmbH, Germany). Direct metal laser sintering was performed using the Yb-fibre laser system EOSINT M270 at regular operating parameters. To investigate the effect of the build direction, the location of the specimens on the CNC platform, and post-heat treatment on the mechanical properties and microstructure development, three groups of DMLS-processed specimens (Group S) were prepared by a variation of sintering parameters:Group S1—the specimens manufactured with different building angular orientations of 0°, 15° and 30° relative to the CNC platform (Figure 2a).Group S2—the specimens built in a 0° direction, but with different positions and locations on the CNC platform (Figure 2b). After laser sintering the specimens of group S1 and S2 were annealed at 750 °C/1 h, according to a regular manufacturer’s procedure. The specimens were cooled down slowly to room temperature. As the microscopic studies did not show that different positions and locations on the CNC platform affect the direction of grain extension and the fatigue crack propagation path, it was decided not to set them in Group S1 and Group S3.Group S3—all specimens built in the 0° direction. The specimens were post-heat-treated in a furnace under an Ar atmosphere at 850 °C/45 min, 880 °C/1 h and 1100 °C/30 min. After post-heat treatment, the specimens (heated at 880 °C and 1100 °C) were cooled down slowly in the furnace until a temperature of 600 °C was achieved, then the furnace door was opened. The specimens post-heat-treated at 850 °C/45 min in the furnace under Ar atmosphere were cooled rapidly [16].

### 2.2. Microstructural Characterisation

Microstructural characterisation was performed on selected places of the prepared tensile specimens: as-cast (C), DMLS-processed with 0° build direction (S1) and post-heat-treated at 1100 °C (S3). The selection of samples was based on the assumption that they represented the best examples of their group. The samples were taken in the part of the tensile specimen where the clamps were, and where the largest diameter was, and the cross-section was examined. In this way, the influence of examining the unstable microstructure resulting from faster cooling was minimised, which is characteristic of thin sections during solidification. With this approach, we wanted to obtain as realistic an assessment as possible of the state of the microstructure and representative samples for evaluation. The metallographic preparation of samples included grinding with SiC paper and polishing with a C suspension. This was followed by chemical etching of the samples. The chemical etchant, 13 g of FeCl_3,_ was dissolved in 40 mL of HCl, and 1 mL of HNO_3_ was added when FeCl_3_ was fully dissolved. The etchant was applied with a dropper (1–3 drops) to the metallographically prepared surface and washed with ethanol after 2–3 s.

The microstructures of the samples were investigated using an optical metallographic microscope, NIKON Epiphot 300 (Tokyo, Japan), with an Olympus DP12 camera (Boston, MA, USA). For detailed microstructure observation and microchemical analyses, a scanning electron microscope, Sirion 400 NC (FEI, Hillsboro, OR, USA), was used, with an energy-dispersive X-ray spectroscopy detector INCA 350 (Oxford Instruments, Abingdon, UK). The additional analysis of phases was performed on selected samples (C, S1, S3) with an XRD Panalytical XPERT Pro PW 3040/60 goniometer 2 theta 10–90° with a step of 0.002° and a time of 100 ms per step. The anode was Cu (Kalfa = 0.154 nm) with a current of 40 mA and a voltage of 45 kV.

### 2.3. Mechanical Properties

Uniaxial tensile testing was performed on a Shimadzu universal tensile/compression testing machine (AG-X Plus, 250 kN, Kyoto, Japan), using standard tensile specimens 3 mm in diameter and a gauge length of 18 mm. At least three (up to 6) tensile specimens for each condition were tested, including as-cast (C), DMLS manufactured and post-heat-treated (S) specimens. The tensile testing was conducted at room temperature with a cross-head speed of v = 2 mm/min (at an initial strain rate of 1.85 × 10^−3^/s). The yield strength (0.2% YS), ultimate tensile strength (UTS) and elongation (El) of the specimens were obtained from the resulting nominal stress–strain curves. The elongations were also determined with the fractured pieces of the specimens fitted together.

After the tensile testing, the fractured surfaces of the specimens were observed using a scanning electron microscope (JEOL-JSM-6610LV, Tokyo, Japan) equipped with an EDS detector (Oxford instruments).

## 3. Results and Discussion

### 3.1. Microstructural Characterisation

The microstructural investigation of the selected samples C, S1 and S3 revealed that the production technology influences its formation greatly, and this can be seen clearly in Figure 3, where the optical microstructure of the etched surfaces of the samples is shown.

The microstructure in sample C is typically dendritic, characteristic for the casting process. The dendrites have a uniform size, and there are no defects or any special inclusions (impurities) in the microstructure. From Figure 3a it can be concluded that the solidification was uniform in all directions, as no inhomogeneity was detected. In contrast, the resulting microstructure in samples S1 and S3 was typical for additive manufacturing technologies, as the basic crystal elements are the so-called hills, the height and width of which depend on the DMLS technique and subsequent thermo-mechanical processing (Figure 3b,c). In the case of the S1 sample, the hills were estimated to be lower at a height below 50 μm, while in the case of the S3 sample, they were significantly higher and more concave. This difference can be attributed to post-heat treatment at 1100 °C, which caused the growth of basic crystal grains, and the merging of smaller ones into larger ones with the goal of stabilisation and reaching the highest possible density of this sample. The microstructure of the Co–Cr alloy prepared by DMLS showed a hierarchical microstructure composed of macroscopic features caused by the gradual melting and solidification of small volumes of the input powder, as melt tracks in a section parallel to the carrier plate of the SLM device [17]. Such a microstructure was also found in many other alloys prepared by SLM [18,19,20]. The columnar grain morphology is related to the epitaxial growth of grains from a previously solidified layer. In some cases, it can form cellular microstructures as a result of high cooling rates (up to 100 °C/s). In the description of the Co–Cr microstructural characteristics compared to other alloys, it is possible to observe differences originating from various parameters of the SLM process, which can influence the formation of the microstructure.

In order to obtain even better insight into the resulting microstructure, we performed SEM investigations on polished samples, with the aim to obtain direct insight without the remains of the elements that were present in the etchant, which is especially important for micro-chemical analysis. Figure 4 shows SEM images of the three representative microstructures.

The SEM microstructures were comparable to the optical ones, although, in some places, the orientation can be detected more precisely, especially in the cases of samples S1 and S3. The results of the microchemical analysis are shown in Table 2. For each sample, 3 × 6 analyses were performed in selected areas, and the average values are shown in Table 2.

A comparison of the measured chemical compositions of all three selected samples shows that sample C had a slightly higher content of Cr, Co and Mo, while the content of W was significantly lower than that of samples S1 and S3. Both S samples had comparable chemical compositions, as the deviation was within the measurement error of the EDX detector. No additional impurities were observed.

XRD analysis revealed the presence of a cubic phase (γ) of Co–Cr [21] and a hexagonal phase (ε) of Co–Cr in all three samples. The diffractograms shown in Figure 5 indicate that the maxima and width of the planes (111)_γ_, (200)_γ_, (20-20)_ε_ and (20-21)_ε_ are different for the studied samples C, S1 and S3. Thus, we can conclude that there are different amounts and distributions of the cubic phase (γ) and hexagonal phase (ε) of Co–Cr in all three samples. These data can also be linked to the chemical composition obtained by EDX analysis (Table 2). In sample C, the most prominent is the plane (111)_γ_, i.e., the cubic phase (γ) of Co–Cr. On the other hand, it can be seen from Table 2 that the composition of Co and Cr was the highest compared to samples S1 and S2. In samples S1 and S2, the plane (20-21)_ε_ is more evident than in sample C, which indicates that the hexagonal phase (ε) of Co–Cr is rather present, and can be related to the influence of Mo and W.

### 3.2. Mechanical Properties

The most important mechanical properties of the specimens from the C and S groups are given in Table 3. The 0.2% YS, UTS and El of C specimens were 687 ± 31 MPa, 827 ± 67 MPa and 8.3 ± 1.8%, respectively. Detailed analysis showed that the DMLS-processed and post-heat-treated specimens showed an enhanced strength level compared to the C specimens. The mean tensile strength of all laser-sintered specimens was increased up to ~1300 MPa, with elongations in the range of 2.7 to 5.3%, depending on the variation in the sintering parameters. Regarding the influence of the build direction of the DMLS specimens (group S1), an increase in the building angular orientation from 0° to 30° severely deteriorated the tensile strength and ductility, as shown in Table 3. The specimens with 0° building direction showed the best combination of strength and ductility, with 0.2% YS, UTS and El of 1255 ± 5 MPa, 1311 ± 40 MPa and 5.3 ± 0.5%, respectively. The 0° specimens were selected to investigate the effect of different locations on the CNC platform (group S2), as well as the influence of post-heat treatment after laser sintering (group S3) on the mechanical properties and microstructure of the Co–Cr alloy.

Tensile testing of the specimens in group S2 showed that there were no significant differences between the strength and ductility of the specimens at different locations and positions on the CNC platform. The mean 0.2% YS was 1184–1190 MPa, the UTS was approximately 1177–1254 MPa, and the El was in the range of 4.2–4.6%, as shown in Table 3.

Two cooling regimes were used in connection with the tensile specimens, which were post-heat-treated (group S3). A slow cooling rate was used after annealing at 880 °C and 1100 °C, while rapid cooling was used for annealing at 850 °C. The results in Table 3 show that all post-heat treatments led to a decrease in the strength compared to the samples of group S1 (0° build direction, annealing at 750 °C). The UTS was decreased from ~1311 ± 40 MPa (sample S1-0°) to ~1145–1183 MPa (group S3). No significant difference in ductility was observed for the samples annealed at 1100 °C (slow cooling rate) and 850 °C (fast cooling rate), although heat treatment at 880 °C (slow cooling rate) caused a drop in the elongation compared to the S1-0° state.

This can also be observed in Figure 6, which displays the stress–strain curves of the as-cast and DMLS-prepared specimens from Co–Cr alloys.

It can be seen from Figure 6 that the highest strength corresponded to the S1-0° state, then to the post-heat-treated samples (S3), while the as-cast state showed the lowest strength level. The slope of the stress–strain curves in the range of elastic deformation was much smaller for the as-cast state than for the DMLS laser-sintered states. Post-heat treatment at 1100 °C provided a slight increase in the slope compared to samples annealed at 850 °C, 880 °C and 750 °C. The observed variations in mechanical properties and stress–strain curve levels can be related to the microstructure characteristics of the as-cast (C) and laser-sintered (S1 and S3) specimens. The superior strength and higher flow stress of the DMLS specimens (S1 and S3) compared to the as-cast specimen (C), can be attributed to a finer microstructure which consisted of small hills developed during local melting and rapid solidification (Figure 3b and Figure 4b). As they became coarser and concave after heat treatment at 1100 °C (Figure 3c and Figure 4c), the strength and flow stress levels were decreased (Figure 6). In addition, the SEM observations shown in Figure 4b,c indicated a difference in the substructure within the hills developed in the as-built (S1) and annealed (S3) specimens. It is likely that a substructure within the small hills contributed to the higher strength level of the S1 specimen. However, a restoration or recrystallisation of the substructure, which is expected to occur during annealing at 1100 °C, was accompanied by a decrease in strength and flow stress. The effect of laser sintering on the microstructure evolution and mechanical properties, as well as the influence of post-heat treatment, were in agreement with the results reported in the literature [22,23,24,25,26,27]. Moreover, the different microstructure characteristics affect the fracture surface appearance after the tensile test, as illustrated by the SEM observations of the as-cast and laser-sintered specimens (Figure 7).

In the case of the as-cast state, the microstructure of the fracture surface showed a distinct dendritic structure, which was characterised by the presence of non-equilibrium solidification, implying inhomogeneity of the chemical composition, a higher probability of the presence of defects, and, consequently, worse mechanical and other properties. In the case of sample S3, which was produced using the DMLS technique and subsequently annealed at T = 1100 °C, we can see that the fracture surface was fairly homogeneous.

Given that the discussed Co–Cr dental alloys are intended for the RPD metal framework in general practice, it should be noted that adaptation by bending is often necessary when handling these alloys. In some cases, such RPDs can break due to small plastic deformation that occurs during dental laboratory work. A greater proclivity for brittle fracture or collapse can be problematic in dental applications. SEM investigations of the fracture surfaces (Figure 7) revealed that the fracture in specimen S3 was more brittle than the fracture in specimen C, which was previously confirmed by the measured total elongation in the tensile failure test. The reasons for these differences can be explained by studying the microstructure and analysing the fracture mechanisms. The crack that occurred due to loading propagated more slowly in the dendritic microstructure, which resulted in a higher toughness of this fracture. Fractographic analysis of the surfaces also revealed that the DMLS fractures were very uniform, with the presence of notches below 1 µm, originating from the presence of sub-micrometre pores in the output DMLS microstructure. This phenomenon is known from the literature as a consequence of binding polymer leakage during the sintering process of metal powder blocks, resulting in brittle fracture [28]. Based on the findings, it can be concluded that the main issue in Co–Cr casting is the unpredictable nature of the process and the increased risk of defects during penetration of the refractory mould and solidification of the alloy during casting, leading to deformation of the framework due to shrinkage [29] at lower 0.2%YS, thus increasing the risk of premature fracture and rework of the prosthesis despite greater toughness.

On this basis, the S3 sample is associated with significantly better measured mechanical properties (see the stress–strain curve in Figure 6), which are important for subsequent dental use. On the other hand, XRD analysis revealed the presence of a cubic phase (γ) [21] and a hexagonal phase (ε) of Co–Cr in all three samples, which meant that the structure of all three investigated samples was similar. Therefore, the manufacturing technology has the greatest influence on the microstructure refinement and final mechanical properties of the dental object. Our findings are in a good agreement with the results reported in the literature [22,23,28,29,30].

Laser melting and sintering of dental Co–Cr alloys provides RPD frameworks with good mechanical properties compared to conventional casting of Co–Cr alloys. Our clinical experience and literature data also show that RPD frameworks made by DMLS fit very precisely on the supporting tissues of partially edentulous patients [31,32]. The future of this technology is warranted by the fact that it is an environmentally friendly technology (medical and municipal solid waste are reduced to a minimum) [33,34,35].

## 4. Conclusions

The following scientific conclusions can be drawn from this study of laser-melted and -sintered dental Co–Cr alloys, intended for producing RPD frameworks:The microstructure of a classically as-cast Co–Cr alloy was dendritic, while the microstructure created using the DMLS technique was typical for additive manufacturing technologies, as the basic crystal elements were the so-called hills.The chemical composition of the samples in the study was comparable for both technologies used, and no additional impurities were observed.XRD analysis revealed the presence of a cubic phase (γ) of Co–Cr [14] and a hexagonal phase (ε) of Co–Cr [15] in all investigated samples.The microstructure of the as-cast fracture surface showed a distinct dendritic structure, while the DMLS surface was homogeneous in grain size and fracture.The superior strength and higher flow stress of the DMLS samples, compared to the as-cast state, can be attributed to a finer microstructure developed during local melting and rapid solidification, as well the varying ratio of γ–Co and ε–Co phases.The tensile properties were also affected by a build orientation, and the highest value of strength was achieved in a 0° direction after laser sintering and annealing at 750 °C. Post-heat treatment at 1100 °C revealed the possibility for achieving a good combination of strength and ductility of Co–Cr dental materials.

Due to the precision of the obtained RPD metal frameworks and the environmentally friendly technology, DMLS technology has a great advantage for the future compared to casting technology.

## Figures and Tables

**Figure 1 materials-16-03267-f001:**
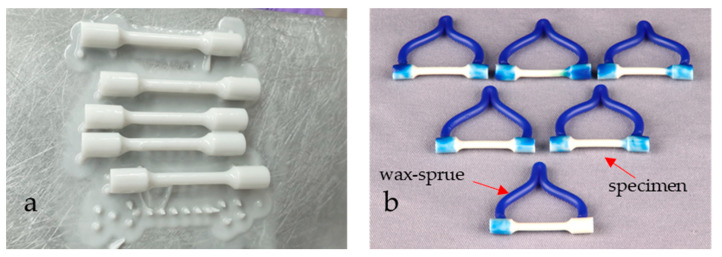
(**a**) Presentation of stereolithography process of the sixth specimen model in photopolymer (3D Dental, Rapid Model Resin, Monocure 3D); (**b**) test specimen model of the control group prepared for investing in the investment material.

**Figure 2 materials-16-03267-f002:**
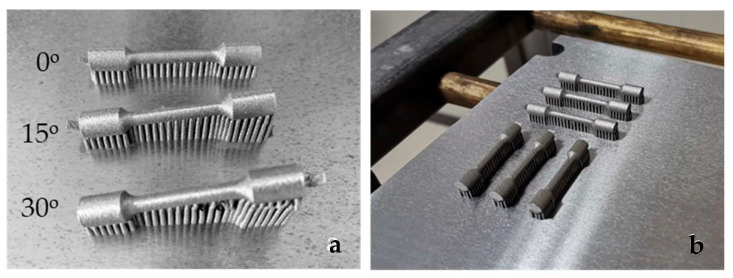
(**a**) S1 group specimens on the CNC machine platform immediately after sintering and before cutting the supports; (**b**) S2 group specimens on the CNC machine platform immediately after sintering and before cutting the supports.

**Figure 3 materials-16-03267-f003:**
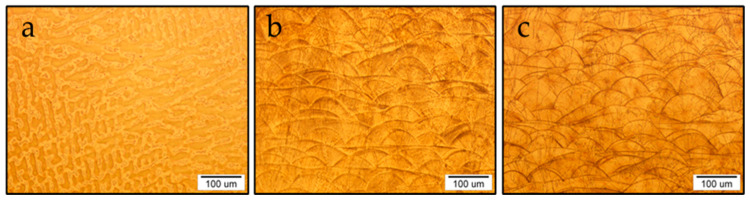
Optical microstructure of samples: (**a**) C, (**b**) S1 and (**c**) S3.

**Figure 4 materials-16-03267-f004:**
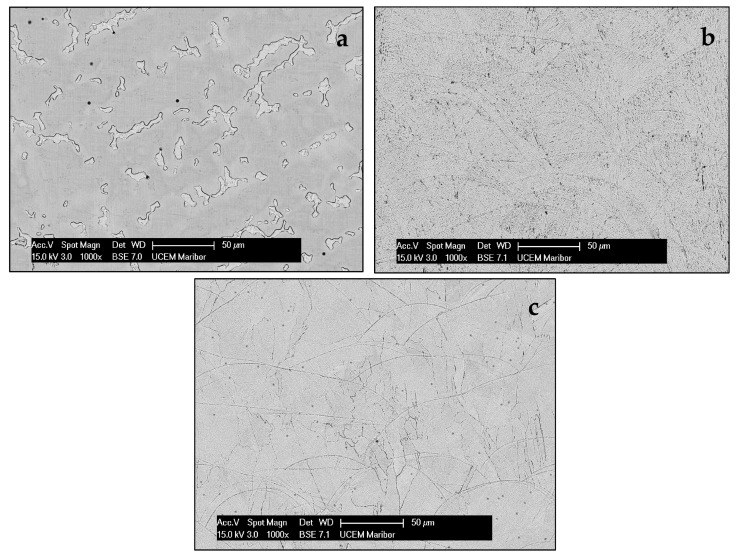
SEM microstructure of samples (**a**) C, (**b**) S1 and (**c**) S3.

**Figure 5 materials-16-03267-f005:**
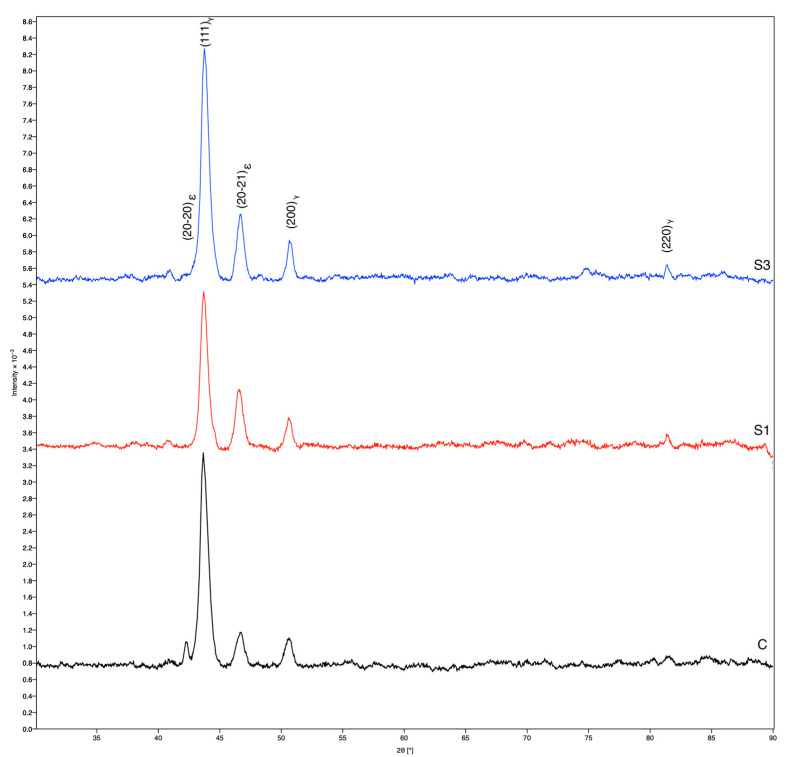
XRD spectra of samples: C, S1 and S3.

**Figure 6 materials-16-03267-f006:**
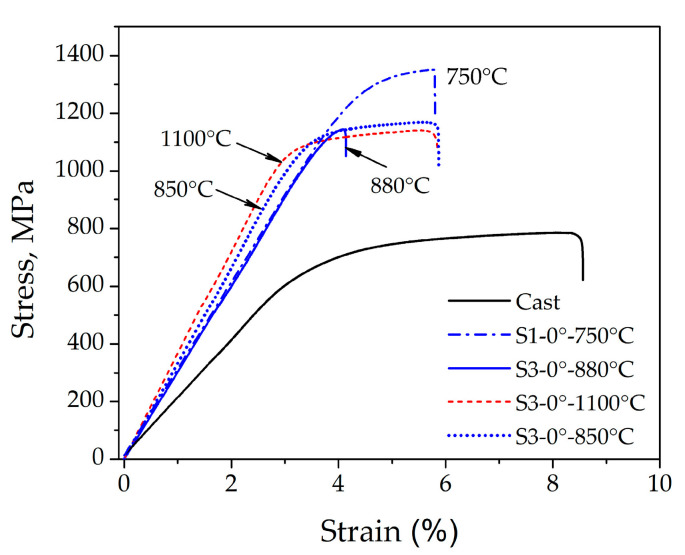
Stress–strain curves of Co–Cr alloy specimens from the C and S groups.

**Figure 7 materials-16-03267-f007:**
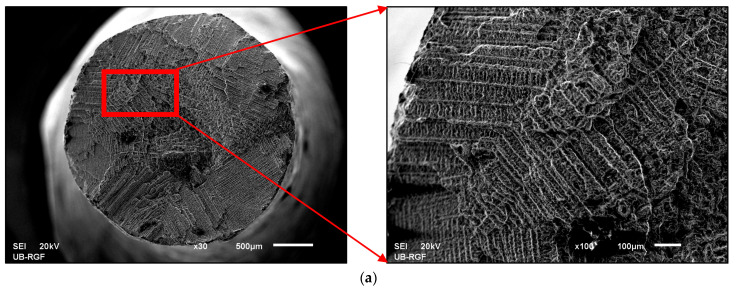
SEM microstructure of fracture surfaces of samples (**a**) C and (**b**) S3 (macro- and micro-view).

**Table 1 materials-16-03267-t001:** Nominal chemical composition (in wt. %) of the used Co–Cr alloys.

Alloy	Co–Cr Alloy for Casting	Co–Cr Alloy for DMLS
Manufacturer	Dentaurum GmbH & Co. KG Remanium GM800+	EOS GmbH—Electro Optical Systems EOS CobaltChromium SP2
Composition		
Co	58.3	62–66
Cr	32	24–26
Mo	6.5	4–6
W	1.5	4–6
Si	1.0	0.8–1.5
Mn	n/a	max 1.5
Fe	n/a	max 0.7

**Table 2 materials-16-03267-t002:** EDX chemical composition in (wt. %) for selected samples.

Sample/wt. %	Cr	Co	Mo	W
C	30.83	60.42	6.28	2.48
S1	24.79	57.42	4.62	13.17
S2	25.12	57.46	4.57	12.85

**Table 3 materials-16-03267-t003:** Tensile properties of as-cast and DM laser-sintered samples of Co–Cr alloys (mean ± SD).

Manufacturing Process	Specimen		0.2% YS, MPa	UTS, MPa	El, %
Conventional Casting	Group C		687 ± 31	827 ± 67	8.3 ± 1.8
Direct Metal Laser Sintering	Group S1	0°	1255 ± 5	1311 ± 40	5.3 ± 0.5
		15°	1225 ± 9	1255 ± 9	4.6 ± 0.1
		30°	/	989 ± 7	2.7 ± 0.3
	Group S2	0°—CNC position 1	1188 ± 3	1254 ± 16	4.6 ± 0.9
		0°—CNC position 2	1184 ± 18	1230 ± 39	4.2 ± 0.8
		0°—CNC position 3	1190 ± 14	1177 ± 69	4.3 ± 1.6
	Group S3	0°—880 °C/1 h	1134 ± 6	1145 ± 15	4.1 ± 0.1
		0°—1100 °C/30 min	1121 ± 52	1182 ± 41	5.1 ± 0.7
		0°—850 °C/45 min	1090 ± 13	1183 ± 14	5.2 ± 0.7

## Data Availability

Not applicable.

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
