# Peer review of "Comparative Study of the Microstructure and Properties of Cast-Fabricated and 3D-Printed Laser-Sintered Co–Cr Alloys for Removable Partial Denture Frameworks"

_materials, 2023, doi:10.3390/ma16083267_

Round 1
Reviewer 1 Report
1. The diffractograms shown in Fig. 6 must be deciphered and analyzed. It can be seen that the intensity of the maxima and their width are different for the studied samples. This effect carries valuable information, but there is no analysis.
2. I would like to note that the authors do not professionally describe the results of the metallographic analysis presented in fig. 4. First, it is not correct to conclude that there are no defects based on the results of optical microscopy. Secondly, when describing the microstructures in fig. 4 b, c the authors use terminology that is not inherent in metallography, for example «basic crystal elements», «hills».
3. It is not clear how the authors could determine the "height" of a structural element from an optical photograph?
4. It is also necessary to correct the description of the results of the fractographic analysis (Fig. 8). Because there is a discrepancy between the explanation of the structure of fractures presented at magnifications of x30 and x100 times.
5. As follows from the comments, the analysis of the results presented in the "Discussion" and "Conclusions" sections requires correction.
Author Response
REVIEW 1
- The diffractograms shown in Fig. 6 must be deciphered and analyzed. It can be seen that the intensity of the maxima and their width are different for the studied samples. This effect carries valuable information, but there is no analysis.
The diffractograms shown in Fig.6 indicate that the maxima and width of the planes (111)γ , (200)γ , (20-20)ε and (20-21)ε are different for the studied samples C, S1 and S3. Thus we can conclude that there are different amounts and distribution of cubic phase (g) CoCr and hexagonal phase (e) CoCr in all three samples. These data can also be linked to the chemical composition, EDX analysis, Table 2.
In sample C, the most prominent is the plane (111)γ , i.e. cubic phase (g) CoCr. From the other side, it can be seen from the Table 2, that the composition of Co and Cr is the highest compared to samples S1 and S2.
In samples S1 and S3, plane (20-21)ε is more evident than in sample C, that indicates that hexagonal phase (e) CoCr is rather present and can be related to the influence of Mo and W.
Text has been added to the article.
- I would like to note that the authors do not professionally describe the results of the metallographic analysis presented in fig. 4. First, it is not correct to conclude that there are no defects based on the results of optical microscopy. Secondly, when describing the microstructures in fig. 4 b, c the authors use terminology that is not inherent in metallography, for example «basic crystal elements», «hills».
The microstructure in sample C is typically dendritic, characteristic for the casting process. So, from Figure 4a it can be concluded that the solidification was uniform in all directions. In contrast, the resulting microstructures in samples S1 and S3 were typical for Additive Manufacturing technologies which depend on the DMLS technique and subsequent thermo-mechanical processing. In the case of the S1 sample, the so-called 'hills' were estimated to be lower while, in the case of the S3 sample, they were significantly higher and more concave. This difference can be attributed to post heat treatment at 1100 °C, which caused the growth of basic crystal grains, and the merging of smaller ones into larger ones with the goal of stabilisation and reaching the highest possible density of this sample.
® as described in the article, a more extensive discussion has been added.
- It is not clear how the authors could determine the "height" of a structural element from an optical photograph?
In Additive Manufacturing technologies the basic crystal elements are usually called 'hills', with appropriate height and width, but which is not a correct explanation from the point of view of metallography.
- It is also necessary to correct the description of the results of the fractographic analysis (Fig. 8). Because there is a discrepancy between the explanation of the structure of fractures presented at magnifications of x30 and x100 times.
The display in the picture has been corrected - text has been added.
- As follows from the comments, the analysis of the results presented in the "Discussion" and "Conclusions" sections requires correction.
Text corrections have been made - all highlighted in red in the manuscript.
Reviewer 2 Report
The manuscript compares the microstructure and mechanical properties of 3-D printed laser-melted and sintered Co-Cr alloys produced at different technological parameters with castings from the same Co-Cr alloy. Some points have been raised and are described below:
1. Line134-135: What’s the point in preparation of Group S2 with different positions and locations on the CNC platform, which was almost not involved in the latter test ? If the purpose is for the setting of repeat samples, the preparation of different positions and locations should be set in Group S1 and Group S3.
2. Line 243-244: The percentage of content of each phases of Co-Cr alloy had better be listed as the proportion of area of the absorption peak , which has an effect on the mechanical properties of the alloy.
3. Line 132: Groupe S1 was divided into specimens with different building angular orientations of 0 °, 15 ° and 30 ° related to the CNC platform, but only one specimen was involved in the latter microstructural characterization tests. Why?
4. Line 301: The units of the horizontal coordinate need to be indicated in Figure 7.
5. Line 307: The annealing temperature “750℃” had better be indicated in Figure 7, otherwise it’s confusing when the “750℃” only appeared in “ sample preparation ”section.
6. Line 310: Why the hill-like microstructure is a “finer” microstructure? The extrapolation process or literature source needs to be elaborated.
7. Line 315: What’s the reason or the mechanism of which the superior strength and higher flow stress can be attributed to the hill-like microstructure? Are there any papers or experiments to support this conclusion?
8. Line 318: The difference of SEM microstructure of fracture surfaces is not only associated with the “decrease of strength and flow stress ”, but also with the ductile and brittle types of failures, which can be indicated in the text.
9. Line 377-379: The “mechanical properties” only takes a small part in the conclusions section, which takes up a large portion of the article.
Author Response
REVIEW 2
The manuscript compares the microstructure and mechanical properties of 3-D printed laser-melted and sintered Co-Cr alloys produced at different technological parameters with castings from the same Co-Cr alloy. Some points have been raised and are described below:
- Line134-135: What’s the point in preparation of Group S2 with different positions and locations on the CNC platform, which was almost not involved in the latter test ? If the purpose is for the setting of repeat samples, the preparation of different positions and locations should be set in Group S1 and Group S3.
The point of preparing group S2 with different positions and locations on the CNC platform was to determine the direction of grain growth and the influence of the resulting sintered microstructure on the path of fatigue crack propagation. Following the observation that microscopic studies did not show that different positions and locations on the CNC platform affect the direction of grain elongation and the path of fatigue crack propagation, it was decided that different positions and locations would not be studied in groups S1 and S3 as in group S2.
- Line 243-244: The percentage of content of each phases of Co-Cr alloy had better be listed as the proportion of area of the absorption peak , which has an effect on the mechanical properties of the alloy.
In the practice of using EDS - (Energy Dispersive Spectroscopy), which emits the entire characteristic X-ray radiation, for the case when there are several chemical elements in the excited sample, the scattered X-ray radiation consists of quanta of those energies that belong to the emission lines of these elements. By measuring the individual quanta of this X-ray spectrum, a chemical analysis of elements with an atomic number higher than beryllium is carried out. The great advantage of EDS is that all incident energy quanta are processed at the same time, and semi-quantitative analysis of elements in small samples is established nowadays. Of course, this method has limitations, such as instability of the system, the influence of accelerating voltage, contamination and errors arising from the calculation of the intensity of pure elements (worse resolution of energies and poorer quantification for low concentrations of elements). Despite the above, the method is very often used and combined with other analytical techniques, so that it provides additional insight into the chemical composition of the sample. All quantification processes of modern EDS analysis are automated within the SEM microscope. Therefore, it is practically impossible for the calculation of the chemical composition according to the provided comment to be carried out on the basis of the area fraction of the absorption peak, because the INCA 350 system itself (Oxford Instruments, UK) is calibrated industrially and as such cannot be changed. Here, it is only necessary to mention that by directing a focused electron beam onto a selected small area of the sample - in our case, the surface analysis was carried out on an area of approx. 25 µm × 25 µm - we obtain the quantitative chemical composition of this area with a depth of approx. 5 µm, which may represent an additional risk regarding the real chemical composition depending on the phase structure.
We added the text, but did not change the results - the table 2.
- Line 132: Groupe S1 was divided into specimens with different building angular orientations of 0 °, 15 ° and 30 ° related to the CNC platform, but only one specimen was involved in the latter microstructural characterization tests. Why?
Only one specimen (building angular orientation of 0o related to the CNC platform) was involved in the microstructural characterization tests because that specimen had the best combination of strength and ductility.
- Line 301: The units of the horizontal coordinate need to be indicated in Figure 7.
There was no units of the horizontal coordinate in Fig. 7, because the Strain values (shown as x-axis) were calculated as a relation between the elongation (mm) and an initial gauge length (mm). Units can be inserted in Fig.7 in case the Strain values are multiplied by 100x. We did the re-calculation of x-axis values and the Strain is now expressed as a relative percentage deformation (%). The scale of x-axis was changed and shown in the range of 0-10 %, instead of 0-0.1 in the previous version. Corrected Fig. 7 is inserted in the revised version of the manuscript.
|
|
|
|
Fig. 7 |
Corrected Fig. 7 – x-axis was changed |
- Line 307: The annealing temperature “750℃” had better be indicated in Figure 7, otherwise it’s confusing when the “750℃” only appeared in “ sample preparation ”section.
Temperature 750°C is indicated in a corrected Fig. 7.
- Line 310: Why the hill-like microstructure is a “finer” microstructure? The extrapolation process or literature source needs to be elaborated.
We came to conclusion that ”finer microstructure which consisted of small hills developed during the rapid melting and solidification (Fig. 4b and Fig. 5b)” based on our microscopic examinations, and we found confirmation in the articles: Zhou, Y.et al. [16], Okazaki, Y. et al. [17], Hitzler, L. et al. [20], Yanjin, L. et al [21], Chen, J. et al [22] and other authors.
- Line 315: What’s the reason or the mechanism of which the superior strength and higher flow stress can be attributed to the hill-like microstructure? Are there any papers or experiments to support this conclusion?
Direct metal laser sintering (DMLS) technology provides fine microstructure development due to repeated local melting and rapid solidification processes within small volume (layer by layer) in a very short time. The grain size are smaller and more homogeneous than in as-cast structure. Rapid solidification decreases dendritic segregation compared to as-cast structure and improves the strength by grain refinement strengthening mechanisms. This was reported in the literature and refs.16-22 were cited in the manuscript. It was also reported in the literature that superior strength can be ascribed to a high dislocation density and a presence of a cell-like substructure within “hills”. Post heat treatment can cause a partial or fully recrystallization providing a possibility for good combination of strength and ductility to be achieved (refs.29-30).
- Line 318: The difference of SEM microstructure of fracture surfaces is not only associated with the “decrease of strength and flow stress ”, but also with the ductile and brittle types of failures, which can be indicated in the text.
Thanks for the advice, the text has been improved and inserted as intended.
- Line 377-379: The “mechanical properties” only takes a small part in the conclusions section, which takes up a large portion of the article.
Conclusion section related to the mechanical properties was revised according to the referees’ suggestions and inserted in a revised version of manuscript.

Reviewer 3 Report
In the article “Comparative Study of the Microstructure and Properties of 2 Cast-fabricated and 3-D Printing Laser-sintered Co-Cr Alloys 3 for Removable Partial Dentures` Frameworks” the authors evaluate the microstructure and mechanical properties of 3-D printed laser melted and sintered Co-Cr alloys, and to perform a comparative study with Co-Cr castings for the same dental purposes.
· In the Introduction part, the author make a short introduction to the DPR field (removable partial denture metal framework), highlighting the advantages of this method but also its limitation with technological development. New additive technologies method, such as stereolithography, selective laser melting and sintering (SLM), direct metal laser sintered (DMLS), 3-D printing, fabrication by lamination) are exemplified with new bibliographic references.
· The aim of the articles is well underline as a comparison between the microstructure and mechanical properties of 3-D printed laser-melted and sintered Co-Cr alloys produced at different technological parameters.
· I consider that the novelty degree of the present articles should be emphasized, i.e. at what research stage the proposed materials are, relative to the literature. Is the 3-D printed Co-Cr alloys already synthesized, are they developed as potential material for dental restorative application, have the specific properties of this kind of material (biocompatibility, mechanical and chemical stability…)?
· The Materials and Methods section well describes the sample preparation methodology, the microstructural and mechanical properties characterization.
· In figure 5, the scale is very small, it must be visible.
· All characterization performed (Microstructural characterization, mechanical properties, tensile proprerties are well structured and conducted with clear aim to demonstrating that the 3-D printed method is viable to the preparation of DPR and does not bring negative properties to the material.
· The conclusion are well organized and in line with the purpose of the articles
· Personally, I consider that some phrase is necessary, to highlight the novelty of the research carried out.
Author Response
In the article “Comparative Study of the Microstructure and Properties of 2 Cast-fabricated and 3-D Printing Laser-sintered Co-Cr Alloys 3 for Removable Partial Dentures` Frameworks” the authors evaluate the microstructure and mechanical properties of 3-D printed laser melted and sintered Co-Cr alloys, and to perform a comparative study with Co-Cr castings for the same dental purposes.
- In the Introduction part, the author make a short introduction to the RPD field (removable partial denture metal framework), highlighting the advantages of this method but also its limitation with technological development. New additive technologies method, such as stereolithography, selective laser melting and sintering (SLM), direct metal laser sintered (DMLS), 3-D printing, fabrication by lamination) are exemplified with new bibliographic references.
Required text related to stereolithography and 3 new references were added.
- The aim of the articles is well underline as a comparison between the microstructure and mechanical properties of 3-D printed laser-melted and sintered Co-Cr alloys produced at different technological parameters.
- I consider that the novelty degree of the present articles should be emphasized, i.e. at what research stage the proposed materials are, relative to the literature. Is the 3-D printed Co-Cr alloys already synthesized, are they developed as potential material for dental restorative application, have the specific properties of this kind of material (biocompatibility, mechanical and chemical stability…)?
In the introduction of the article (Line 63-64) it was pointed out that the sintered Co-Cr alloy is acceptable for fixed dental restorations - restorations of small dimensions (all parts of the restoration are rigid). RPD framework is a more demanding dental restoration from a mechanical point of view, some parts of this restoration are rigid, and some are elastic (Line 66-68). For this reason, the optimal protocol (alloy sintering parameters) for making the RPD framework is still being sought. The sintered Co-Cr alloy is biocompatible and chemically stable in the oral cavity.
Co-Cr dental alloys, which are the subject of scientific treatment, are already present in general dental practice, individual articles also separately describe the characteristics and properties of these alloys, which are produced using different techniques - see references from [1-6]. However, no systematic comparison of the Co-Cr alloys shown in this scientific discussion has been found anywhere in the literature. At the end of the Introduction, a sentence was added that emphasizes the importance of the dependence of the properties of Co-Cr alloys on the manufacturing technology.
- The Materials and Methods section well describes the sample preparation methodology, the microstructural and mechanical properties characterization.
- In figure 5, the scale is very small, it must be visible. The improvement was made.
- All characterization performed (Microstructural characterization, mechanical properties, tensile proprerties are well structured and conducted with clear aim to demonstrating that the 3-D printed method is viable to the preparation of DPR and does not bring negative properties to the material.
- The conclusion are well organized and in line with the purpose of the articles
- Personally, I consider that some phrase is necessary, to highlight the novelty of the research carried out.
We added the text at the end of the Introduction: The purpose of this article was to demonstrate the wider scientific public that there are differences in microstructure and thus consequently in properties of CoCr dental alloys, despite identical raw material output, due to different manufacturing and subsequent heat treatments technology, with which we address the novelty of this research work.

Round 2
Reviewer 1 Report
It's nice that the authors responded to all the comments. After the corrections and additions made, the article can be published.